# Erythro–Magneto–HA–Virosome: A Bio-Inspired Drug Delivery System for Active Targeting of Drugs in the Lungs

**DOI:** 10.3390/ijms23179893

**Published:** 2022-08-31

**Authors:** Alessio Vizzoca, Gioia Lucarini, Elisabetta Tognoni, Selene Tognarelli, Leonardo Ricotti, Lisa Gherardini, Gualtiero Pelosi, Mario Pellegrino, Arianna Menciassi, Settimio Grimaldi, Caterina Cinti

**Affiliations:** 1Institute of Organic Synthesis and Photoreactivity (ISOF), National Research Council of Italy, Via Gobetti 101, 40129 Bologna, Italy; 2The BioRobotics Institute, Scuola Superiore Sant’Anna, Piazza Martiri della Libertà 33, 56127 Pisa, Italy; 3Department of Excellence in Robotics & AI, Scuola Superiore Sant’Anna, Piazza Martiri della Libertà 33, 56127 Pisa, Italy; 4National Institute of Optics (INO), National Research Council of Italy, Via G Moruzzi 1, 56124 Pisa, Italy; 5Institute of Clinical Physiology (IFC), National Research Council of Italy, Via G Moruzzi 1, 56124 Pisa, Italy; 6Institute of Translational Pharmacology (IFT), National Research Council of Italy, Via Fosso del Cavaliere 100, 00133 Rome, Italy

**Keywords:** engineered erythrocytes, cell-based drug delivery systems, pulmonary drug delivery, active targeting, magnetic platform

## Abstract

Over the past few decades, finding more efficient and selective administration routes has gained significant attention due to its crucial role in the bioavailability, absorption rate and pharmacokinetics of therapeutic substances. The pulmonary delivery of drugs has become an attractive target of scientific and biomedical interest in the health care research area, as the lung, thanks to its high permeability and large absorptive surface area and good blood supply, is capable of absorbing pharmaceuticals either for local deposition or for systemic delivery. Nevertheless, the pulmonary drug delivery is relatively complex, and strategies to mitigate the effects of mechanical, chemical and immunological barriers are required. Herein, engineered erythrocytes, the Erythro–Magneto–Hemagglutinin (HA)–virosomes (EMHVs), are used as a novel strategy for efficiently delivering drugs to the lungs. EMHV bio-based carriers exploit the physical properties of magnetic nanoparticles to achieve effective targeting after their intravenous injection thanks to an external magnetic field. In addition, the presence of hemagglutinin fusion proteins on EMHVs’ membrane allows the DDS to anchor and fuse with the target tissue and locally release the therapeutic compound. Our results on the biomechanical and biophysical properties of EMHVs, such as the membrane robustness and deformability and the high magnetic susceptibility, as well as their in vivo biodistribution, highlight that this bio-inspired DDS is a promising platform for the controlled and lung-targeting delivery of drugs, and represents a valuable alternative to inhalation therapy to fulfill unmet clinical needs.

## 1. Introduction

In the last decades, pharmaceutical technology has witnessed a remarkable advancement in the development of new solutions capable of overcoming the solubility/stability, bioavailability and administration limitations of several drugs, and the improvement of the efficiency/safety profiles of old and new drugs by means of target-based technologies. Equally significant attention has been devoted to identifying the most suitable route of administration to optimize the dosing and effectiveness of target therapies designed for reaching hardly accessible districts, such as the internal organs. In this contest, the lungs have been the cause of intensive study, mostly focusing on the possibility of delivery therapy by the pulmonary route, a noninvasive administration for the systemic and local delivery of therapeutic agents. This strategy takes advantage of the physiological lung’s gas and small molecule exchange potential, sustained by an intense pulmonary capillary blood flow, of which the high permeability and large absorptive surface area, as well as the thin epithelial cell layer, contribute to create an ideal environment for drug delivery and absorption [1]. Despite the ideal environmental conditions, treating lung parenchyma in situ could be difficult when a locally effective high concentration therapy is needed, for example, in cancer [2,3] or fibrosis [4]. On the other hand, systemic treatment is often ineffective due to the efficient clearance by the phagocytic cells of the liver, spleen and lung [5].

As for the intravenous administration of nanoparticles, the nature of the material as well as the size, shape and surface properties, are critical parameters for inhalation therapy, since they can affect the pharmacokinetics and biodistribution of the carried drug [6,7,8,9,10,11,12], thus impairing the treatment efficacy. In fact, the respiratory tract has the ability to effectively remove pathogens and particulates through the action of beating cilia, mucus, immune cells, transporters and enzymes [6], making local drug delivery to the lungs challenging [13,14,15]. Thus, there is an urgent need to find a novel and effective drug delivery system (DDS) that efficiently crosses the physical and biological lung barriers and minimizes the clinical and technical gaps.

Herein, the feasibility of using an erythrocyte-based DDS for noninvasive drug delivery in the lung has been explored. Red blood cells (RBC), called erythrocytes, are naturally occurring biological vesicles which are an alternative to their synthetic counterparts, featured by a large and substantial, but reversible, elastic deformability in any arbitrary shape, which is a crucial element to reach any body’s organs/tissues through efficient navigation into the microcirculation. Multiple animal and human studies have shown the feasibility of drug encapsulation into the RBC and coupling to the RBC surface [16]. However, the loading procedures represent a sticky step which could affect the integrity, plasticity, mechanical robustness and deformability of the RBC membrane. This could lead to undesired effects, such as an uncontrolled release of the drug in the plasma and damage/senescence of membranes, triggering erythrocytes accumulation exclusively in the liver and the spleen, with the consequent elimination through macrophage phagocytosis in a few days [17,18,19].

Based on this, while RBC DDS could serves as depot for the continuous or prolonged systemic administration of therapeutics in cases where the controlled release of the cargo is not required [20,21], it results poorly effective in those clinical settings entailing tissue/organ-targeted and specific drugs’ administration. The controlled and targeted release of drugs at the site of action from RBC carriers is a challenging and fascinating goal, and innovative solutions are needed to circumvent these issues.

In this view, we have designed and patented [22] a novel erythrocyte-based DDS, the Erythro–Magneto–Hemagglutinin (HA)–Virosomes (EMHVs), engineered to simultaneously carry magnetic nanoparticles and therapeutic compounds within their inner volume and endowed with filamentous hemagglutinin fusion proteins on their membranes. This EMHV platform combines the advantages of natural cellular entities for systemic circulation, the physical properties of synthetic magnetic nanoparticles, enabling a selective targeting under external local magnetic field, and a biofunctional virus-inspired infection/docking mechanism for controlled drug release in targeted tissues/cells. We previously demonstrated that the EMHVs have the ability to encapsulate a wide variety of therapeutic compounds (chemical and biological agents), protect them from biodegradation and increase their stability and bioavailability in vivo [23,24,25,26,27]. Furthermore, because of their biofunctional characteristics, we proved that the EMHVs are able to fuse with and release their cargo into the host cells, mimicking the viral infection mechanism [23].

Here, the potential use of the EMHV as an efficient, safe and noninvasive bio-inspired functional DDS for the local treatment of pulmonary diseases has been investigated. The mechanical and biophysical properties of EMHVs, such as the membrane elasticity and deformability levels, osmotic resistance and magnetic susceptibility, have been explored to gather information on their resilience to the mechanical stress which could be met in the capillary bed, as well as on the level of responsiveness to the magnetic field necessary to retain a large number of EMHVs in a selected area of the lung. The proof of concept on the ability of EMHVs to cross the physical and biological lung barriers, and on the efficacious local nanocarriers concentration and membrane interaction and fusion in the lung selected areas, under the action of a defined external magnetic field, has also been evaluated in vivo indicating the EMHV as a therapeutic tool for the precise and effective therapeutic intervention in the lung.

## 2. Results

### 2.1. Biomechanical and Biophysical Properties of EMHVs

In the first part of this study, the biomechanical and biophysical properties of EMHVs were characterized. It is known that the elasticity of erythrocytes allows them to travel through the microvascular bed by deforming their body into a fuse-like shape and then restoring in a biconcave fashion. When erythrocytes stiffen, because of aging or some pathologies, they risk being destroyed by the mechanical stress encountered while traveling through the narrow vessels [28].

Therefore, the elasticity of the EMHVs has been studied in vitro in comparison to elasticity of naïve RBCs in order to obtain indications on their resilience after injection in the circulatory system. Scanning ion conductance microscopy (SICM) was used to measure the EMHV membrane’s elasticity. SICM uses a glass pipette filled with a saline solution as a probe to obtain topographic images with submicrometric resolution without the contact of cells adhered to the substrate [29,30]. SICM operation is based on the measure of the ionic current flowing across the pipette tip when a bias voltage is applied. The current decreases when the distance between the probe tip and the nearest insulating surface is reduced. Thus, the current value is used to control the probe–sample distance during scanning. This technique can be useful to investigate cell elasticity by applying controlled pressure to the probe in order to produce a solution flow through its tip aperture, which in turn produces a local deformation of the underlying cell [31,32]. In particular, the response of the erythrocyte membrane to the pressure applied via a SICM pipette has been recently described [33]. The EMHVs deposited in the petri dish, in contact with a poly-l-lysine (PLL) coating, adhered to the substrate, assuming a dome shape similarly to normal RBCs, as shown in Figure 1a,b by the representative SICM topographical maps. For the dome-shaped EMHVs, the diameter was ~8 µm and the height was ~2.5 µm. We measured the indentation caused by a pressure of 500 Pa applied above the cell center in the normal RBCs (control group, n = 95, from two healthy donors) and EMHVs (n = 131, from the same donors). The histograms in Figure 1c,d show the distribution of the values obtained. It can be noticed that the median indentation extent under the application of a pressure of 500 Pa is significantly larger in the EMHVs (97 nm) than in the control RBCs (35 nm). We also compared the RBCs and EMHVs indentation distributions on a single donor basis. Figure 1e shows a box chart where the indentation observed for the EMHVs has been normalized over the median indentation of the “naïve erythrocytes” (normal RBC) of the corresponding donor. As the engineering process of the EMHVs is composed of several steps, in which cells undergo different treatments, we therefore investigated the separate effects of these treatments on the biomechanical properties. The results are shown in Figure 1f. These data refer to a third healthy donor, and the indentation values are normalized over the median value observed for the normal RBCs (control group, n = 20) of this donor. The cells that underwent the only lysis/resealing process (L/R, n = 42), those which resealed only with hemagglutinin (FHA, n = 31) and those only with magnetic nanoparticles (NP30, n = 20) were all more compliant than the normal RBCs. These results indicate that the lysis/resealing process per se is sufficient to increase the membrane indentation in response to mechano-stimulation.

The single-cell-based SICM method allows estimating the local membrane deformability, thus the whole-cell globular resistance to osmotic stress was also measured. It is known that the circulating erythrocytes (normal RBC) are embedded in plasma with an osmolarity of 300 mOsM [34]. Therefore, it is possible to suspend the cells in increasing hypotonic solutions and observe at which osmolarity value the lysis and the consequent hemoglobin release occurs. Because of the low sensitivity of this method, the osmotic resistance of the normal RBCs and the engineered erythrocytes (EMHV) did not differ significantly (data not shown), suggesting that the engineering processes do not sensitize the EMHVs to damage by osmotic stress.

To estimate the magnetic force F_m_ acting on the EMHVs capable of guaranteeing an efficient docking effect at the site of action by an external permanent magnet, the volumetric magnetic susceptibility *χ_v_* and magnetization M were calculated. In particular, the magnetization grade in response to the magnetic field was estimated, considering that the force by which the EMHVs are attracted by a permanent magnet, in addition to the characteristics of the magnet itself, depends on their volumetric magnetic susceptibility and their residual magnetization. To determine these parameters, dried samples of superparamagnetic iron oxide nanoparticles (SPION25-30, Nanocs Inc., Boston, MA, USA) and EMHVs were analyzed through a vibrating sample magnetometer (VSM). Figure 2a shows the magnetization (in emu g^−1^) of the superparamagnetic nanoparticles. Figure 2b shows the average magnetization of 0.127 × 10^9^ EMHVs containing 3.98 ng of magnetic nanoparticles in response to magnetic field (Oe). The volumetric magnetic susceptibility *χ_v_* of the EMHVs was calculated by determining the slope of the line that best approximates each set of EMHV magnetization data (emu g^−1^) in response to the magnetic field (Oe) in the region between 200 Oe and 600 Oe (Figure 2c).

These values of the magnetic field have been chosen based on electromagnetic simulations carried out by using the COMSOL Multiphysics software, modeling the magnets typically used in in vivo tests. Results showed that these magnets produced a magnetic field of 355 Oe at a distance of 2 mm from the surface, corresponding to the distance between the mouse lung and the magnet.

The slope of the line that best approximates each data series in this magnetic field region represents the mass magnetic susceptibility (*χ_m,CGS_*) of the EMHVs, expressed with the CGS system measurement units: (𝑒𝑚𝑢 𝑔^−1^ 𝑂𝑒^−1^). Using equations reported in the experimental section, the mass magnetic susceptibility can be converted into the volumetric magnetic susceptibility (*χ_ν,SI_*), which is dimensionless. For 0.127 × 10^9^ EMHVs, the volumetric magnetic susceptibility is *χ_ν,SI_* = 2.089 × 10^−4^ ± 0.862 × 10^−4^, suggesting that the EMHVs have high magnetic susceptibility, and thus can be remotely controlled by external magnetic fields after their systemic intravenous injection.

### 2.2. Delivery of EMHVs in the Lung

In the second part of this study, we highlighted the potential application of EMHVs as drug delivery systems for the local treatment of lung diseases such as lung cancer.

To provide evidence on the efficiency and efficacy of the EMHV drug delivery system to reach and selectively accumulate in the lung, and to prove the feasibility of its application in clinical settings, noninvasive bioanalytical imaging was performed in mice treated with EMHVs. Actually, various imaging modalities, including magnetic resonance imaging (MRI), computed tomography (CT), near-infrared (NIR) imaging and optical immuno-histochemical imaging were applied to visualize the biodistribution of the micro- or nano-sized DDSs in living systems and to monitor the in vivo pharmacokinetics [35,36,37]. Herein, NIR was used to evaluate the biodistribution of EMHVs in the whole animal after 1 h intra-caudal vein injection and the ability to flow through the bloodstream and reach the pulmonary area, where the magnetic field gradient was generated by an external permanent magnet localized and properly held on the back. NIR fluorescence imaging was chosen because it offers higher sensitivity and a better signal-to-background (S/B) ratio compared to visible spectra. More importantly, due to the reduced light absorption and scattering of NIR light in animal tissues, and the low tissue autofluorescence in the NIR region, NIR fluorescence is well suited for in vivo animal imaging [38]. Therefore, 1 × 10^9^ IRDye800CW-labeled EMHVs (Ex/Em: 774/789 nm) were used for in vivo targeted tracking.

As shown in Figure 3a, an efficient inclusion of IRDye-labeled superparamagnetic nanoparticles was detected in almost all engineered erythrocytes (EMHVs). An N52 NdFeB magnet, contained within a silicon backpack (Figure 3b) positioned on the back of the animal in correspondence of the lung area (Figure 3c), was applied for 30 min soon after the intravenous administration of the IRDye800CW-labeled EMHVs to promote the accumulation of this magnetic DDS in the pulmonary area. An enrichment of the IRDye signal in the area of the ribcage, with a low off-target interaction with other regions of the animal body, were observed (Figure 3d), suggesting an efficient selectivity of the EMHV accumulation in the area where the magnetic field was generated.

To verify the nature of the NIR signal, the magnetic resonance T_2_-weighted images (MRI) and, in particular, the mean and the median T_2_ relaxation time values were investigated on the treated animals and controls.

MRI has become widely used as a tool for noninvasive clinical diagnoses, exploiting various inorganic nanoparticles administered before the scanning to increase contrast. The contrast agents used for the MRI analysis are generally based on iron oxide nanoparticles, which produce a signal decreasing effect and provide hypointense signals in T_2_-weighted images in shorter time spin–echo sequences [39]. Therefore, the presence of superparamagnetic iron oxide nanoparticles (SPION) derived from the EMHVs interaction with cells in the pulmonary area was successfully detected by the MRI. The T_2_ relaxation time signals, evaluated pixel per pixel in the identified regions of interest (ROI) of the lung, were evaluated.

As shown in Figure 4, in which a representative coronal area level is reported as an example, a general reduction of T_2_ spin–spin relaxation signals was evident from the lung map and confirmed in all the area levels in the EMHV-treated mouse (Figure 4b) with respect to untreated one (Figure 4a). The hypointense signals measured in the EMHV-treated mouse suggest the accumulation of magnetic nanoparticles delivered by the EMHVs in the lung. In fact, both the mean and the median of the T_2_ relaxation time values, evaluated on the total lung map reconstruction, were calculated in the EMHV-treated mice with respect to untreated one (Mean: 40.12 vs. 46.16 ms; Median: 34.44 vs. 39.31 ms), showing a reduction of more than 13%. Furthermore, it has been described that damaged/senescent erythrocytes [17] and magnetic nanoparticles, when administered intravenously, have a short plasma half-life and quickly accumulate in the liver [40], which are then degraded. To exclude possible unwanted tissue interaction and content leaking, the T_2_ spin–spin relaxation signals have also been evaluated on the total liver map reconstruction of the treated animals and controls. In Figure 4c,d, the representative central coronal areas of the livers of the treated and untreated animals have been reported. No difference in the relaxation time of both the untreated (c) and EMHV-treated mice (d) was measured in the liver, suggesting that no off-target EMHV accumulation occurred (see also Appendix A).

The efficient selectivity of EMHV delivery in the lung area, where the magnetic field was applied, was supported by finite element model (FEM) simulations, showing the magnetic properties, i.e., magnetic flux density and magnetic field streamlines, in the surrounding area of the permanent magnet. As shown in Figure 5, the magnetic FEM simulations indicate that the maximum magnetic flux density (or magnetic field) generated by the external permanent magnet is about 1.31 Tesla (T), and decreases until 0.24 T at a distance of 2 mm from the magnet surface (Figure 5a,b).

Furthermore, observing the magnetic field distribution, the magnetic field is higher immediately below the magnet, which is the physical point where the magnetic field streamlines closes again on the magnet, and it quickly decreases when moving away from the magnet surface. In Figure 5c, we visualized the simulated magnetic flux density curve superimposed, in terms of magnet position and orientation, to an MRI mouse lung area subjected to a higher magnetic field. The magnetic field streamlines mostly overlap with the hypointense areas identified by the MRI imaging, suggesting the efficiency of the magnetic source in localizing and capturing the systemically injected EMHVs in the lung area.

As a further proof of the magnetically induced localization of the EMHV, a postmortem investigation was performed in both the whole specimen and the selected regions of the treated lung.

To verify the achievement of the magnetically driven EMHVs to fuse with and release the cargo at the site of action, the presence of both superparamagnetic iron oxide nanoparticles (SPION), conveyed by the EMHVs, and *bordetella pertussis* filamentous hemagglutinin (FHA) fusion protein, inserted in the EMHV membranes, were highlighted by Perls’ Prussian blue staining and FHA immunostaining, respectively.

Computed tomography (CT) imaging was performed to obtain 3D anatomical information/reference on the explanted paraformaldehyde-fixed lungs. A definitive spatial resolution was described, as the lung offers an excellent soft tissue contrasting capability for the identification of the magnetic EMHV deposit in the pulmonary area. As shown in Figure 6a, a CT attenuation and contrast was mostly observed in the superior lobes of the lung (500–700 μm) with respect to inferior lobes (1100 μm), which was in line with the MRI hypointense signals detected in these areas. Based on this anatomical information, longitudinal sections were obtained at 500, 700 and 1100 μm for the immunostaining analysis. In fact, once docked at the site of interaction, the EMHV presence can be identified through the reactivity for filamentous hemagglutinin (FHA) monoclonal antibodies. Moreover, the presence of super-paramagnetic iron oxide nanoparticles (SPION), conveyed by EMHV, were identified by Perls’ Prussian blue staining.

Representative images of the presence of both iron accumulation (Figure 6b–d) and FHA-positive broncho-alveolar cells (Figure 6e–g) at 500 and 700 μm were shown, indicating the successful anchoring of magnetically-controlled site-specific EMHVs and their fusion with the targeted cells. No Perls’ Prussian blue and immunohistochemical staining were detected in sections at 1100 μm (Appendix A).

## 3. Discussion

Developing carrier systems capable of delivering pharmacologically relevant doses of a therapeutic compound into the desired organ/tissue/cell in a targeted way, and to protect these drugs from the environment and rapid degradation, remains a crucial challenge for a successful therapy.

Externally controllable carriers maximizing the efficacy and safety of a therapy is an exciting paradigm that can be exploited to revolutionize the field of precision nanomedicine [41]. Among the different strategies that can be used to provide a DDS with a good degree of control over drugs’ transport, the use of a magnetic field seems to be one of the most suitable and clinically relevant solutions [42,43].

In this context, the magnetic EMHV carrier appears as one of the most promising candidates for carrier-controlled locomotion within the vasculature, since we demonstrated here and in previous works in cancer mouse models [26,27] that EMHVs can be selectively directed and can release the incorporated content at the site of interest through external magnetic fields. The first prerequisite to such a scope is the ability of the EMHVs to maintain the resilience to the mechanical stress met in the capillary beds. This would avoid the risk of systemic toxicity due to the uncontrolled release of the drug. The strategy used to generate engineered EMHVs is based on the modification of the content and surface molecules of natural erythrocytes empowered with better docking interaction with cells in targeted tissues. Here, we have shown that EMHVs exhibit a mechanical compliance higher than RBCs. This difference is probably due to the partial loss of hemoglobin induced by the temporary formation of pores in the cell membrane. These biomechanical properties make EMHVs flexible enough to effectively navigate into capillaries and reach any vascularized organs such as the lung.

In support of this, the reported the results of the NIR and MR imaging, as well as the immunohistochemical staining, clearly showing that the EMHVs efficiently accumulated in the lung, particularly in the area where the magnetic field was applied, and fused with the target epithelial cells, indicating that the anatomical characteristics of lung (high pulmonary capillary blood flow, thin epithelial cell layer and extensive vascularization) are an ideal environment for EMHV delivery system. Additionally, for the efficient encapsulation of both drug and magnetic nanoparticles within their inner volume, and the insertion of hemagglutinin fusion proteins on their membranes, the calibrated processes customized to transiently open pores at the membrane level without causing hemolysis/rupture of erythrocytes have allowed the production of erythrocyte-based DDS, which maintain the physiological membrane asymmetry and fluidity, thus preventing membrane damage/senescence of the EMHVs, and therefore their sensitization to osmotic and mechanical stress. This maintained physiological properties have circumvented the uncontrolled release of the encapsulated cargo in the plasma, and the accelerated EMHV clearance in the liver, as evidenced by NIR and MRI in vivo imaging, where no signal of magnetic nanoparticles outside the lung were detectable.

The second prerequisite for achieving the scope of an efficient targeted drug delivery at the desired site of action is the magnetic susceptibility of EMHVs, making them controllable by external magnetic fields. The results of our study on the biophysical properties of EMHV carriers and FEM magnetic simulations highlight that they have a high degree of magnetic susceptibility and can be remotely controlled by external magnetic fields for local lung delivery. Our research fit well with the increasing interest of the literature in the field of magnetic force for noninvasive interventions and treatments. The 3D reconstruction of the simulated magnetic flux density distribution on MR lung imaging indicates the capability of a magnetic source to localize and capture EMHV carriers in the lung anatomical site. In fact, with respect to the original erythrocytes, the SPIONs content and the membrane-inserted FHA epitopes favor, respectively, the localization and membrane interaction and fusion with the tissue cells under the influence of the magnetic field. We demonstrated, by using live imaging, the fast effect by the magnet that drives the accumulation at the lung sites. As well, the postmortem analysis by the immuno-reactivity of the FHA and the iron element clearly shows that the EMHVs selectively accumulated in those areas of the lung under the influence of the magnetic field, while in those adjacent areas not influenced by the magnetic force the presence of the EMHV was not detectable. Holding the carriers at the site significantly increases the rate of interaction and fusion with the epithelial cells, where the high pulmonary capillary blood flow, and the thin epithelial cell layer and extensive vascularization are an ideal environment for EMHV delivery system. We also demonstrated that the process for obtaining EMHVs from natural erythrocytes, using the patented protocol, allows for the yield of robust carriers. Red blood cells undergo customized transient opening of the membrane pores without causing hemolysis/rupture. This would allow the possibility of introducing active molecules into the vessels but preventing membrane damage/senescence of the EMHVs and sensitization to osmotic and mechanical stress. This retained physiological profile circumvents the uncontrolled release of the encapsulated cargo in the plasma and shield the EMHVs from clearance in the liver, as demonstrated by the NIR and MRI in vivo imaging, where no signal of magnetic nanoparticles were detectable.

The controlled locomotion and viral-inspired biological properties make this intravascular triggered drug delivery system a bio-based functional microrobot with the ability to locally deliver and release the carried drug at the site of action. As we recently showed [44], a properly designed device able to generate a nonuniform magnetic field can optimize the therapeutic selectivity in more restricted and focalized areas of interest, rendering EMHV platform a promising “trojan horse” for delivering therapeutic drugs to specifically selected target zones, thus enhancing patients’ adherence to the treatment and minimizing the complications which could derive by the systemic administered of free drugs. Furthermore, here we show that the injected EMHVs can provide the real-time imaging and detection of their biodistribution by proper visualization methods, such as NIR and MRI, rendering this cell-based delivery system a fascinating theranostic tool applicable in the clinical setting.

## 4. Materials and Methods

### 4.1. Preparation of IRDye-Labeled EMHVs and Optical Imaging

Human erythrocytes were isolated from the whole blood samples of healthy donors by gradient centrifugation at 400× *g* for 30 min and then washed twice in 1x PBS. Informed consent of all donors were obtained in accordance with European guidelines and approval (no. PG0092248/2019) of the Ethical Committee. The engineered erythrocytes (Erythro–Magneto–HA–Virosomes, EMHV) were prepared following the patent protocol (patent no. WO/2010/070620). Briefly, 2 × 10^9^ erythrocytes were partially open in lysis buffer to remove hemoglobin, and the isotonicity was then restored by adding resealing buffer supplemented with lyophilized *Bordetella pertussis* filamentous hemagglutinin (FHA) fusion protein (2 μg) (Sigma Aldrich, St. Louis, MO, USA) and 25–30 nm ammine functional superparamagnetic iron oxide nanoparticles (SPION25-30) (0.1 mg) (Nanocs Inc., Boston, MA, USA) labeled with IRDye 800CW-NHS ester near-infrared (NI) dye according to manufacturer’s protocols (LI-COR biosciences, Cambridge, UK) for in vitro and in vivo imaging. The suspension was incubated for 45 min at 37 °C under mild agitation to promote resealing and obtain IRDye-labeled engineered erythrocytes (IRDye-EMHV). To confirm the inclusion of IRDye-labeled superparamagnetic nanoparticles, an Olympus IX81 Inverted microscope system equipped with a halogen bulb (Olympus, Hamburg, Germany) was used. NIR filters (EX: 710/75 nm, EM: 810/90 nm; Chroma Technology Corp., Rockingham, VT, USA) were used for IRDye 800CW detection.

### 4.2. Scanning Ion Conductance Microscopy (SICM) Analysis for EMHV Membrane Elasticity

Both normal (RBC) and engineered erythrocytes (EMHVs) from the same donor were stored (hematocrit 20%) in cryovials at 4 °C in physiological solution (PS) containing: NaCl (155 mM), KCl (5 mM), Hepes (10 mM), pH 7.4 and supplemented with bovine albumin (1 mg mL^−1^).

For the SICM measurements, cell suspension (330 µL) was diluted in PS (2 mL), centrifuged at 500× *g* for 5 min and resuspended in 2 mL of PS three times. Polystyrene petri dishes were prepared by placing a drop of poly-l-lysine (PLL) (0.1 mg mL^−1^) on the bottom surface and incubating for 1 h at room temperature (20 °C). Then, excess PLL was rinsed out using PS. PLL-coated petri dishes were filled with PS (4 mL) with CaCl2 (1 mM) and MgCl2 (1 mM). Cell suspension (20 µL) was dropped in the bath solution. At contact with the PLL coating, the cells adhered, assuming a dome shape [28]. The experiments were performed at room temperature. The specimen prepared as described was used for a maximum of 4 h. SICM measurements started within 24 h from preparation of EMHVs and ended by 108 h after preparation. Compliance measurements were carried out in the same experimental conditions on both engineered and naïve erythrocytes from the same donor. Pipettes were prepared from borosilicate capillaries (1/0.5 mm outer/inner diameter, with filament) by means of a BB-CH-PC puller (Mecanex, Geneve, Switzerland). The pipettes were filled at capillary equilibrium [45] with PS with bivalent ions. Pipette resistance was in the range 28–32 MΩ. The back of the pipette was locked to a pressure-tight holder by means of a silicon O-ring. The holder was connected via silicone tubing to a syringe driven by a micrometric screw for pressure regulation and to a pressure transducer (FPM Fujikura, Tokio, Japan) for monitoring. The position of the probe tip was controlled by a step motor for coarse movements along the z axis and by a linearized piezo translation stage Nanocube P-611 3S, with a driver E-664 (Physik Instrumentes GmbH, Rosenheim, Germany) for fine adjustments and scan. The pipette tip was positioned above the target cell through the help of an inverted optical microscope Nikon Diaphot 300 (Nikon Instruments, Amstelveen, NL), equipped with a 40× Ph3 objective and a CCD camera. Then, in order to optimize the reproducibility of the measurement, we used a fast scan SICM procedure to center the pipette on the top of the cell [33]. SICM measurements were carried out in back-step mode [46], where the probe position and the ionic current were simultaneously recorded while the pipette was lowered toward the cell at a constant speed (1 µm s^−1^) until a given percent of current reduction was reached; then, the pipette was moved back to the initial height. The pressure value to be applied in the measurements was chosen based on two main constraints: (i) the pressure should be low enough to avoid damages to the cell and (ii) the resulting indentation should be a small fraction of the local cell height [47]. Therefore, after a preliminary check, we devised to use a pressure of 500 Pa. Isolated cells were chosen for compliance measurement. For each cell, approaching curves were recorded at ambient pressure (9 repetitions) and then applied an over a pressure of 500 Pa to the pipette (9 repetitions). Nonparametric Kruskal–Wallis one-way analysis of variance on ranks test was used with the significance of *p* < 0.05.

### 4.3. Osmotic Resistance Test

Erythrocytes (1.8 × 10^8^) or EMHVs were added to increasingly hypotonic NaCl solutions (1 mL): (in mOsm) 300 (isotonic), 200, 150, 100, 0 (H2O). Each cell suspension was mixed gently and then centrifuged at 400× *g* for 2 min. The presence of hemoglobin in the supernatant indicated cell lysis.

### 4.4. Magnetization Grade and Response to Magnetic Field of EMHVs

A total of 1.27 × 10^8^ EMHVs, engineered with unlabeled superparamagnetic iron oxide nanoparticles (SPION25-30) (Nanocs Inc. Boston, MA, USA), were analyzed by a VSM (PPMS 6000) (Quantum Design Ltd., Stevenage, UK) to estimate the amount of nanoparticles entrapped into the EMHVs and the magnetic force acting on the EMHV DDS. After the engineering processes, all the EMHV samples were washed four times in physiological phosphate buffer saline (PBS), pH 7.4, by centrifugation at 400× *g* for 10 min at 4 °C to remove all the nanoparticles that were not incorporated into the EMHVs. Different independent samples were dehydrated and tested. The EMHV (1.27 × 10^8^, 500 µL) sample and nanoparticles (5 mg mL^−1^, 50 µL) were frozen at −80 °C in a teflon cup for 2 h and then completely dried by a freeze-dryer for 20 h. The dried samples were finally encapsulated in a teflon pill, weighed and analyzed with the VSM at room temperature. The volumetric magnetic susceptibility *χ_ν_* of the EMHVs was calculated by determining the slope of the line that best approximates each set of EMHV magnetization data (emu g^−1^) in response to the magnetic field (Oe) in the region between 200 Oe and 600 Oe.

The slope of the line that best approximates each data series represents the mass magnetic susceptibility (*χ_m,CGS_*) of the EMHVs, expressed with the CGS system measurement units: (𝑒𝑚𝑢 𝑔^−1^ 𝑂𝑒^−1^). This value can be converted into the unit of measurement of the international system (*χ_m,SI_*), as follows:(1)$$\chi_{m,\;SI}\left(\frac{cm^{3}}{g}\right)=\chi_{m,\;CGS}\;\left(\frac{emu}{g}\right)\;4\pi \;\left(\frac{cm^{3}}{emu}\right)\;$$

To obtain the volumetric magnetic susceptibility *χ_ν,SI_*, which is dimensionless, it is necessary to multiply the value of *χ_m,SI_* for the density of the sample:(2)$$\chi_{v,\;SI}=\chi_{m,\;SI\;}\left(\frac{cm^{3}}{g}\right)\rho_{EMHV}\;\left(\frac{g}{cm^{3}}\right)\;$$

The density of the EMHVs was considered analogous to that of normal erythrocytes, as it is assumed that the addition of nanoparticles, which have negligible volume and mass compared to those of an erythrocyte, and the loss of part of the proteins contained, do not appreciably alter the mass of the EMHVs. Furthermore, previous experiments have shown that there are no conspicuous differences in the diameter of the EMHV compared to that of the erythrocytes. Considering that, the density value used is: *ρ_EMHV_* = 1.1 g cm^−3^ [48]. Once the magnetic characterization of the EMHV delivery system (DDS) was performed, it was possible to estimate the magnetic force F_m_ allowing the EMHV DDS anchoring at the target site in presence of a known magnetic field (H) produced by the external magnet by using the following equation:(3)$${\overset{\text{→}}{F}}_{m}=V\overset{\text{→}}{M}\cdot \nabla \mu_{0}\overset{\text{→}}{H}=\mu_{0}V\frac{3\left(\chi_{v,SI}-\chi_{0}\right)}{\left[\left(\chi_{v,\;SI}-\chi_{0}\right)+3\right]}\left(\overset{\text{→}}{H}\cdot \vec{\nabla }\right)\vec{H\;}\;$$
where V is the DDS volume, µ_0_ is the vacuum permeability and *χ* and *χ*_0_ are the magnetic susceptibility of the samples, defined by Equation (3), and the magnetic susceptibility of the vacuum, respectively [49]. To obtain an efficient magnetic anchoring of the EMHV DDS, the constraint described by the following equation was respected:(4)$${\overset{\text{→}}{F}}_{m}+{\overset{\text{→}}{F}}_{D}\geq 0\;$$
where F_D_ is the drag force in the capillary that can be approximated by applying the Stokes law for a sphere in a laminar flow (EMHV DDS can be considered a sphere), as follows
(5)$${\overset{\text{→}}{F}}_{D}=-6\pi \omega r\vec{v}\;$$
where *ω* is the viscosity of the blood, *r* is the radius of the sphere that approximates the EMHV DDS and *v* is the EMHV DDS speed [50]. In order to respect the constraint described by Equation (3), the magnetic field and the magnetic field gradient produced by the magnet (a N52 Neodymium–Iron–Boron (NdFeB), the strongest magnet available in the market, cylinder with a diameter of 5 mm and a height of 10 mm) were estimated through finite element analysis (FEA) simulations. In agreement with the anatomical constraints of the animal model, the magnet was placed at a distance of 2 mm from the target site. The FEM results were preliminarily validated by an experimental assessment carried out by using a Hall sensor, Magnetometer KOSHAVA 5 (Wuntronic, GmbH, Munich, Germany). In order to obtain an efficient anchoring of the EMHV delivery system, by elaborating the Equations (2)–(5), the following condition has to be respected for the magnetization *M*, expressed in A m^−1^:(6)$$M\geq \frac{9\mu v}{2r^{2}\nabla B}$$

### 4.5. In Vivo Animal Model and Treatment

Athymic Balb/c Nude mice were intracaudal vein (i.v.)-injected with 1 × 10^9^ IRDye-labeled EMHVs resuspended in PBS (200 µL). The control group was treated with saline solution (PBS) (200 µL). After the injection, an N52 NdFeB magnet (diameter of 10 mm and height of 6 mm) (Supermagnete GmbH, Gottmadingen, Germany) was applied on the back of the animals in correspondence of the right lung to generate an external static magnetic field. After 30 min of exposure to the local magnetic field, the mice were anesthetized and underwent NIR imaging. After NIR imaging, the animals were sacrificed by CO_2_ asphyxiation and analyzed for magnetic resonance imaging. For preparing whole lung specimen to use in micro-CT and immunohistochemical analysis, intratracheal injection of paraformaldehyde (4%) was used to fix the lungs before the explant. The care and use of the animals used here was strictly carried out in accordance with European guideline Directive 2010/63/EU and National Regulation for the Protection of Vertebrate Animals used for Experimental and other Scientific Purposes in force and approved by the local Animal Welfare Body and the Italian Ministry of Health (CNR-COD-270111). Animal wellbeing was monitored accordingly to the indications of Langford et al. [51].

### 4.6. In Vivo Near-Infrared Imaging of IRDye-Labeled EMHV

Near-infrared (NIR) imaging was used to evaluate the biodistribution of the EMHV: IRDye-labeled EMHVs (1 × 10^9^) diluted in PBS (200 µL) were injected through the intracaudal vein (i.v.). After 30 min of exposure to the local magnetic field on the ribcage area, the mice were anesthetized with isofluorane (2%) and the total body scanning was acquired using the NIR Odyssey imaging instrument (LI-COR biosciences, Cambridge, UK). The Ex/Em settings for the 700 nm channel and 800 nm channel were 685/720 nm and 785/820 nm, respectively. The images were analyzed using the accompanying software.

### 4.7. Ex Vivo Magnetic Resonance Image and Analysis

The ex vivo MRI experiments on the sacrificed mice were performed with a 7 Tesla Pharmascan 70/16 small animal MRI system (Bruker Pharmascan, Ettlingen, Germany) using a circular polarized resonator for mouse whole-body loadings. After a scout acquisition to localize the lung, multislice Turbo-RARE T2-weighted images were acquired for the anatomy of the lung, and MSME and MGE methods were used to measure T_2_ values, respectively. Acquisition parameters for Turbo-RARE were: (i) repetition time (TR) = 2632 ms, (ii) first echo time (TE) = 11 ms, (iii) field of view (FOV) = 5 × 5 cm, (iv) slice thickness = 1 mm, (v) number of slice = 24, (vi) timing = 8 min 25 s. Acquisition parameters for MSME map were: (i) TR = 4621 ms, (ii) TE = 11 ms, (iii) echo train length (ETL) = 15, FOV = 5 × 5 cm, matrix 256 × 256, slice thickness = 1 mm, number of slice = 24, timing = 59 min. Acquisition parameters for MGE map were: TR = 3500 ms, first TE = 3.82 ms, echo spacing = 4.73 ms, number of echoes = 20, flip angle (FA) = 30 deg, FOV = 5 × 5 cm, matrix 256 × 256, slice thickness = 1 mm, timing = 33 min. MRI images were postprocessed to correct bias field in the homogeneity due to the surface receiver coil and filtered with an anisotropic coherence filtering algorithm, and then segmented with a semiautomatic image analysis algorithm adapted from Chan–Vese and developed in-house using the Matlab (The Mathworks, Natick, MA, USA) software suite on a per slice basis, and the total volume was calculated as integral among all identified lung regions [52]. Regions of interest (ROIs) were then manually drawn to identify the lung area in each subject and each slice. Pixel-by-pixel single exponential fits of signal intensity as a function of TE were done to generate transverse (T_2_) relaxation maps using the MSME and MGE sequences.

### 4.8. Simulation of Magnetic Field Distribution on Lung

With the aim of investigating the distribution of the magnetic field produced by the external N52 permanent magnet fixed on the mice’s skin, and thus estimating the areas of the lung most subject to the release of the drug, the FEM COMSOL Multiphysics simulation software (COMSOL, Palo Alto, CA, USA) was used. A dedicated model was implemented with the “Magnetic Fields, No Currents (mfnc)” module of the Comsol AC/DC library. The magnetic field distribution was simulated by including a 5 mm in diameter and 10 mm height NdFeB N52 (remanence flux density of 1.43 T) axial cylindrical magnet within a suitable air medium with a magnetic permeability similar to human tissues. By exploiting the cylindrical shape of the magnets, imposed by the geometrical constraints of the working environment, the geometrical symmetry was used to reduce the computational cost of the simulation. Furthermore, cylindrical air-filled shells around the magnetic units were included in the simulation to refine the mesh resolution, thus improving the simulation results, the calculation capacity and the computational time without making the simulations significantly demanding. The entire geometry was meshed by using a Free Tetrahedral distribution with different sizes. The complete custom mesh consisted of 13,049 and 3144 volume and superficial domains, respectively, and 511 boundary elements for a total of 25,082 degrees of freedoms to be solved.

### 4.9. Immunohistochemical Analysis

Paraformaldehyde (4%) fixed the lungs explanted from the EMHV-treated animals were included in the paraffin-embedded (FFPE) blocks for histology tissue processing. Before immunohistochemical analysis, the samples were examined by computed tomography (CT) imaging to obtain 3D anatomical information/reference. Micro-CT scans were performed with the Xalt-HR scanner, [53] which is a variable-geometry cone-beam CT system with embedded software for geometrical misalignment compensation, equipped with a microfocus X-ray source (20–50 kV) and a digital flat-panel X-ray detector of 2048 × 1024 pixels, having an active area of 10 × 5 cm. Micro-CT scans were performed at 50 kV, 0.7 mA, 2.5 min of total scan time and were reconstructed at an isotropic voxel size of 74 μm using a modified Feldkamp-type cone-beam Filtered Back projection (FBP) algorithm [54]. For the immunohistochemical analysis of the EMHV delivery in the lung, specimens of the explanted lung and included in the paraffin were cut by rotary microtome Microm HM 300 (Bio-Optica, Milano, Italy) at the 500, 700 and 1100 micron longitudinal sections. Perls’ Prussian blue staining were used following standard protocols to detect the presence of ferric iron superparamagnetic nanoparticles in lung. For immunohistochemical filamentous hemagglutinin (FHA) detection, individual slides from FFPE specimens were incubated in a humid chamber overnight at 4 °C with mouse anti-FHA (filamentous hemagglutinin mAb, Statens Serum Institute) at the dilution 1:50. After 5 min washing with PBS Tween (20%), secondary biotinylated antibody (30 min) and Vectastain Elite ABC reagent (30 min) were applied. For detection, slides were incubated with peroxidase substrate solution (DAB), counterstained with standard Meyer Hematoxylin protocol (Sigma-Aldrich, Saint Louis, MO, USA) and analyzed under an Olympus BX43 light microscope interfaced to a DP20 video camera for digital acquisition (Olympus Life Science, Tokyo, Japan). Imaging analysis was carried out by using CellSens Dimension Olympus software.

## 5. Conclusions

In summary, we successfully demonstrated that the Erythro–Magneto–HA–virosomes (EMHV), designed to reach the target tissues/cells by means of an external magnetic field properly held in the right functional position, and to control drug release with a mechanism similar to the viral infection one, showed excellent biomechanical and biophysical properties which have allowed them to efficiently cross the physical and biological lung barriers. The high grade of both membrane deformability and magnetization susceptibility have guaranteed the resilience of the EMHVs to the mechanical stresses met by traveling into narrow vessels, and thereby provide an efficient magnetically-controlled accumulation at the desired site of action after systemic injection. In addition to these properties, the efficiency of the EMHV DDS to fuse with and trigger the release of incorporated “alien” substances into targeted tissue/cells of lung is shown, highlighting the potential of this naturally designed bio-inspired functional DDS as an attractive alternative to other delivery strategies for the systemic delivery of pharmaceuticals in the lungs, therefore improving the therapeutic interventions in this area.

## 6. Patents

Erythrocyte-based delivery system, method of preparation and uses thereof (patent no. WO/2010/070620); Inventors: Caterina Cinti, Antonella Lisi and Settimio Grimaldi; Applicant: Consiglio Nazionale delle Ricerche (IT).

## Figures and Tables

**Figure 1 ijms-23-09893-f001:**
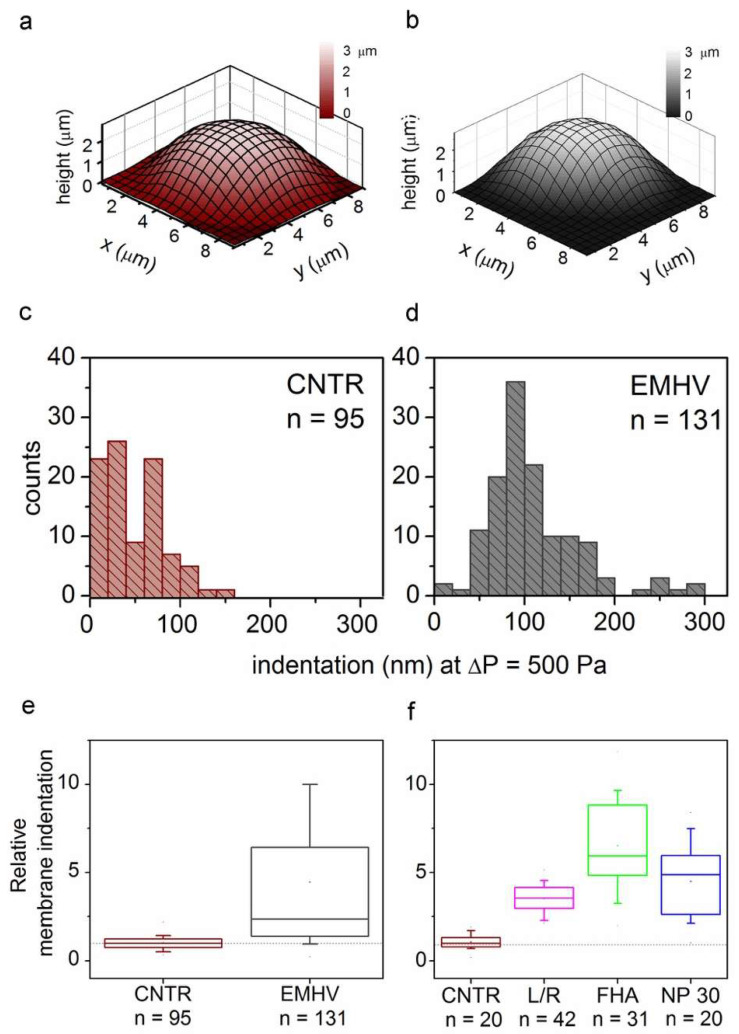
(**a**,**b**) Representative SICM maps of an RBC and an EMHV, respectively, adhered to a PLL-coated petri dish. Maps are obtained with no pressure applied to the pipette, in back step mode, with stopping criterion at 98.5%. Lateral step: 500 nm; (**c**,**d**) Distribution of membrane indentation values in the population of controls (normal RBCs, CNTR, n = 95) and EMHVs (n = 131). Median values are 35 nm and 97 nm, respectively; (**e**) Distribution of relative membrane indentation values (normalized over the median of normal RBCs of the same donor). CNTR as in (**c**) and EMHV as in (**d**). Whiskers show 10–90 percentile range; min, max and mean values are also shown; (**f**) Distribution of relative membrane indentation values (normalized over the median of normal RBCs of the same donor) after the different treatments used to obtain the EMHVs. CNTR: normal RBCs; L/R: lysed and resealed; FHA: lysed and resealed with only hemagglutinin inserted on membrane of erythrocytes; NP30: lysed and resealed with only 30 nm nanoparticles inside the erythrocytes. Whiskers show 10–90 percentile range; min, max and mean values are also shown.

**Figure 2 ijms-23-09893-f002:**
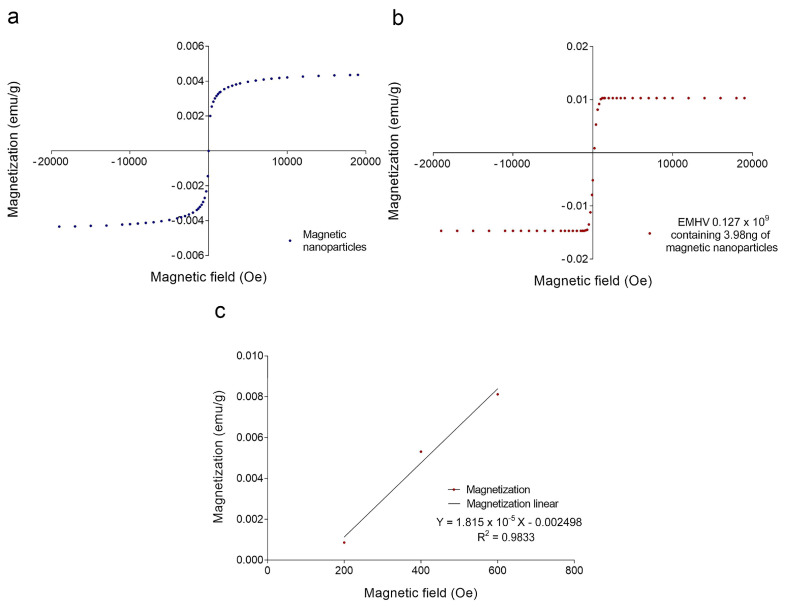
(**a**) Magnetization curve (emu g^−1^) of a dried sample of superparamagnetic nanoparticles in response to a magnetic field (Oe); (**b**) Average magnetization (emu g^−1^) of 0.127 × 10^9^ EMHV dried samples in response to magnetic field (Oe); (**c**) Average magnetization (emu g^−1^) in response to the magnetic field (Oe) in the region of magnetic field between 200 and 600 Oe. The slope of the line that best approximates this data series represents the mass magnetic susceptibility (*χ_m,CGS_*) of the EMHVs.

**Figure 3 ijms-23-09893-f003:**
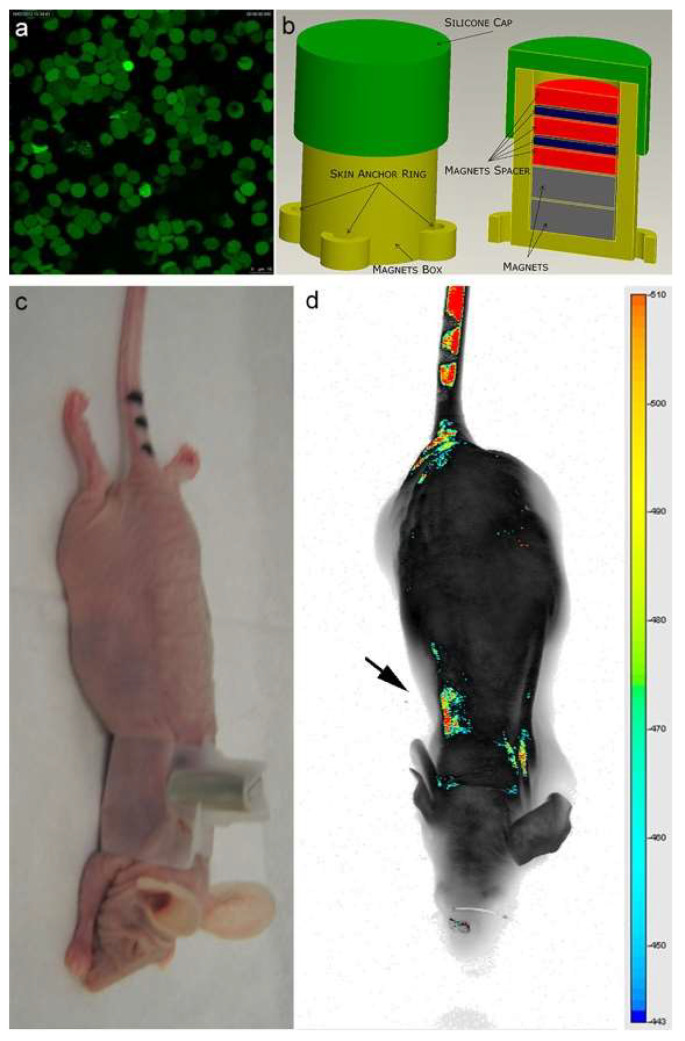
(**a**) Confocal laser scanning microscopy imaging of Erythro–Magneto–HA–Virosomes (EMHVs) incorporating the IRDye800CW-labeled magnetic nanoparticles; (**b**) Schematic representation of silicon backpack containing removable magnet; (**c**) Athymic Balb/c Nude mice with the silicon backpack containing the removable magnet on its right back; (**d**) The longitudinal NIR in vivo imaging (LICOR-Bioscience): dorsal view of IRDye 800CW-labeled EMHVs pseudocolor fluorescence distribution in the whole body. The black arrow indicates the position of the magnet.

**Figure 4 ijms-23-09893-f004:**
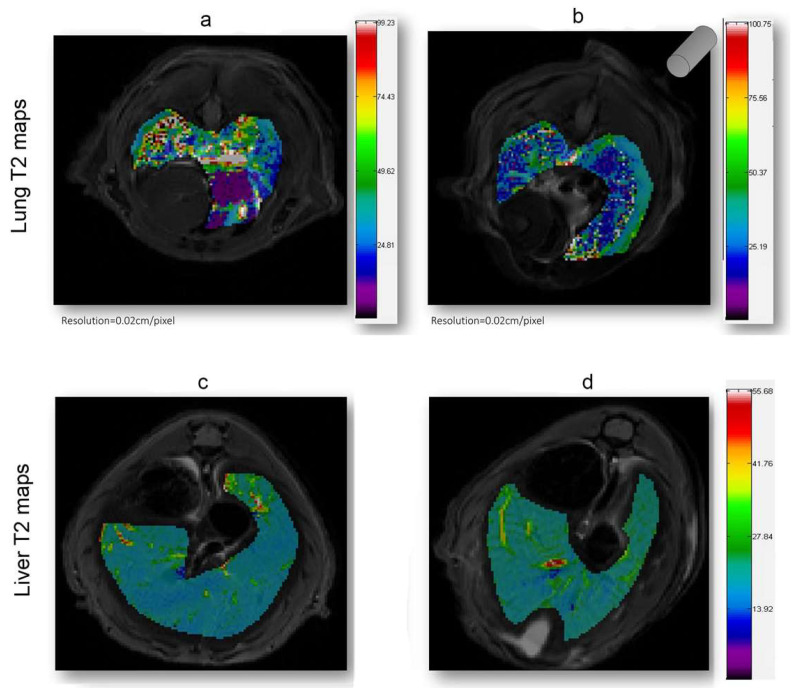
T_2_ (ms) maps in the lung and liver ROI areas. A general reduction of relaxation time in EMHV-treated mouse (**b**) is visible in lung when compared with untreated one (**a**). No difference in relaxation time of both untreated (**c**) and EMHV-treated mice (**d**) is evident in the liver. Map resolution equal to 0.02 cm/pixel. The gray cylinder in (**b**) indicates the position of the magnet.

**Figure 5 ijms-23-09893-f005:**
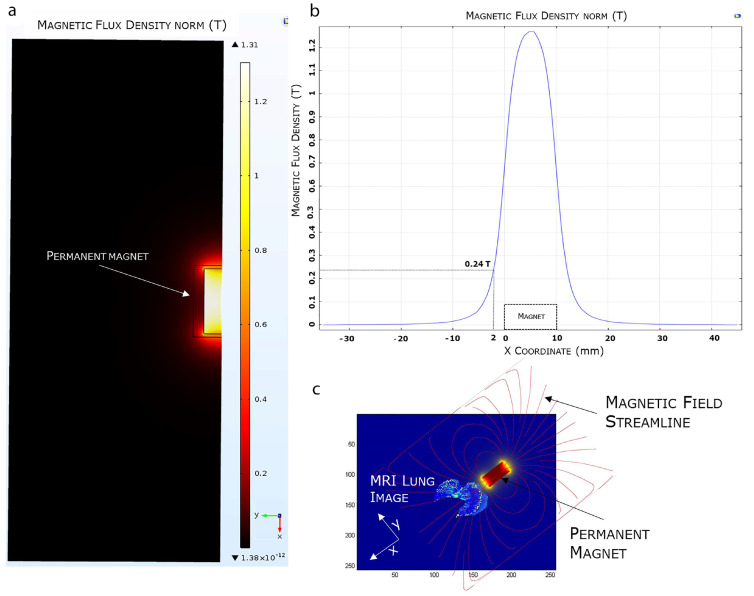
FEA magnetic simulation using COMSOL software. (**a**) 2D representation of the simulated magnetic flux density distribution measured in Tesla (T). (**b**) Magnetic flux density variation. (**c**) Overlapping of the 3D simulated magnetic field on a mice MRI lung model.

**Figure 6 ijms-23-09893-f006:**
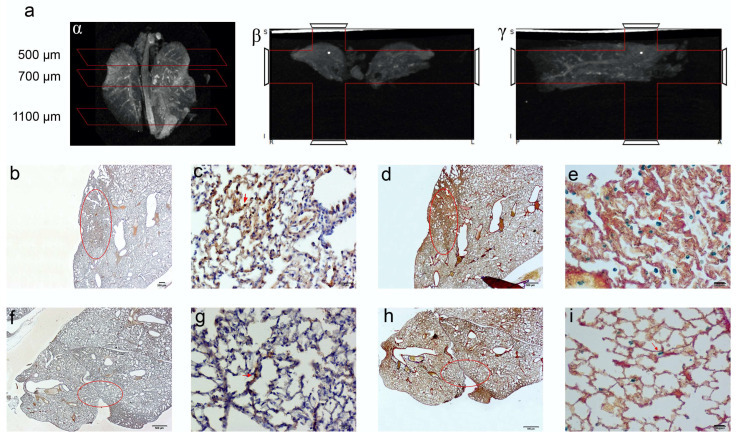
CT imaging of paraformaldehyde fixed lungs embedded in paraffin. The iron enrichment is visible in the corona (**a**) axial (**b**) and sagittal (**g**) plan images between 500 and 700 μm of the lung length due to the presence of EMHV accumulation. Immunohistochemical investigation (**b**–**i**) for the localization of the EMHV by magnetic force application identified by FHA and Spions contents reactivity on consecutive slides. (**b**–**e**) Reactivity in slides at 500 mm level; (**b**) The 20× magnification and (**c**) 40× magnification images of reactive FHA DAB immune-staining. (**d**) The 20× magnification and (**e**) 40× magnification of Perl‘s Prussian Blue reactivity of Iron particles. Less intense signals have been detected outside the magnetic field. (**f**–**i**) Immune reactivity measured at 700 mm level. (**f**) The 20× magnification and (**g**) 40× magnification images of reactive FHA DAB immunostaining. (**h**) The 20× magnification and (**i**) 40× magnification of Perl‘s Prussian Blue reactivity. The red circles in (**b**,**d**,**f**,**h**) correspond to the stained area showed in (**c**,**e**,**g**,**i**) pictures. Absence of reactivity was detected at 1100 mm (Perl’s Prussian Blue staining Appendix A).

## Data Availability

Not applicable.

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
