# Peer review of "Erythro–Magneto–HA–Virosome: A Bio-Inspired Drug Delivery System for Active Targeting of Drugs in the Lungs"

_ijms, 2022, doi:10.3390/ijms23179893_

Round 1

Reviewer 1 Report

This study aims at improving DDS for delivery to the lung vasculature by loading magnetic suspension into RBC ghosts, which should allow their magnetic retention in an area of interest using externa magnetic field. The practical utility of this approach needs clarification since this methodology may work only for delivery to the targets with known localization, and this magnetic retention provides no control over the cellular delivery of the cargoes, which presumably will be removed by blood perfusion immediately after the release from RBC carrier. There are several issues, which must be adequately address.

Major points.

1. The novelty of this work and proposed approach requires substantial consideration. I would address this issue by adding the following paragraph alone with cited references:

RBC-based DDSs are explored for decades (PMID 34597748; 20410503; 15525799; 3462715; 25151978; 25992439; 8867894). Multiple animal and human studies showed feasibility of drug encapsulation into the RBC (PMID 24968029; 22833997; 22230036; 2128736) and coupling to RBC surface (PMID 32397513; 29365311; 27836986; 26228773; 19616049). This study describes a novel multi-component DDS based on loading into RBC magnetic nanoparticles allowing to retain the DDS after intravascular injection in an organ of interest, in the present iteration the pulmonary vasculature.   

2.You must test what happens after intra-arterial injection. IV administration as used in this study may favor retention due to mechanical factors such as micro-aggregation of modified RBC carriers.

3. T2 imaging (e.g., Figure 4) must be complemented by direct measuring of the load accumulating in lungs and liver with and w/o magnetic field.

4.The effect on biocompatibility of carrier RBC must be addressed more rigorously both in vitro and in vivo. Initial naïve RBC and resultant RBC-DDDs must be tested side-by-side for sensitivity of RBC to osmotic lysis, complement, mechanical stress and oxidative stress, as described and in vivo by direct tracing of blood clearance and uptake by RES, lungs, kidneys (PMID 27003833; 29371620).

5.Key parameters of design (size, PDI, charge, content) need to be indicated. What is function of FHA in this study? How many FHA units are per RBC? Which are “the targeted cells” (line 312), and why FHA-RBC supposed to selectively target them?

6. Informative value of tissue sections in figure 6 is low because of poor resolution and tissue characterization. Why the uptake in the sections of different thickness varies so dramatically?

Minor issues

Discussion is too long. Generally, all text and literature related to the respiratory route can be removed.  

Author Response

This study aims at improving DDS for delivery to the lung vasculature by loading magnetic suspension into RBC ghosts, which should allow their magnetic retention in an area of interest using externa magnetic field. The practical utility of this approach needs clarification since this methodology may work only for delivery to the targets with known localization, and this magnetic retention provides no control over the cellular delivery of the cargoes, which presumably will be removed by blood perfusion immediately after the release from RBC carrier. There are several issues, which must be adequately address.

Major points.

  1. The novelty of this work and proposed approach requires substantial consideration. I would address this issue by adding the following paragraph alone with cited references: RBC-based DDSs are explored for decades (PMID 34597748; 20410503; 15525799; 3462715; 25151978; 25992439; 8867894). Multiple animal and human studies showed feasibility of drug encapsulation into the RBC (PMID 24968029; 22833997; 22230036; 2128736) and coupling to RBC surface (PMID 32397513; 29365311; 27836986; 26228773; 19616049). This study describes a novel multi-component DDS based on loading into RBC magnetic nanoparticles allowing to retain the DDS after intravascular injection in an organ of interest, in the present iteration the pulmonary vasculature.

Answer: We received the manuscript as suggested and we added the paragraphs in the Introduction and the Discussion.

2.You must test what happens after intra-arterial injection. IV administration as used in this study may favor retention due to mechanical factors such as micro-aggregation of modified RBC carriers.  

Answer: We acknowledge the concern about the possible toxic consequence of the intravenous injection of EMHV, however in the past we performed effective and safe IV injections of drug loaded and unloaded EMHV to test their potential as DDS for anticancer therapy in xenograft cancer models (Grifantini, R.; Taranta, M.; Gherardini, L.; Naldi, I.; Parri, M.; Grandi, A.; Giannetti, A.; Tombelli, S.; Lucarini, G.; Ricotti, L.; et al. Magnetically driven drug delivery systems improving targeted immunotherapy for colon-rectal cancer. J. Control. Release 2018, 280, 76–86, doi:10.1016/j.jconrel.2018.04.052.; and Naldi, I.; Taranta, M.; Gherardini, L.; Pelosi, G.; Viglione, F.; Grimaldi, S.; Pani, L.; Cinti, C. Novel epigenetic target therapy for prostate cancer: A preclinical study. PLoS One 2014, 9, e98101, doi:10.1371/journal.pone.0098101) and to deliver a genetic therapy to revert cardiac hypertrophy in vivo (Kusmic C, Vizzoca A, Taranta M, Tedeschi L, Gherardini L, Pelosi G, Giannetti A, Tombelli S, Grimaldi S, Baldini F, Domenici C, Trivella MG, Cinti C. Silencing Survivin: a Key Therapeutic Strategy for Cardiac Hypertrophy. J Cardiovasc Transl Res. 2022 Apr;15(2):391-407. doi: 10.1007/s12265-021-10165-1). In all cases the therapy was administered chronically (at least for 4 weeks) with no adverse side effects and no animal death was related to EMHV administration.

  1. T2 imaging (e.g., Figure 4) must be complemented by direct measuring of the load accumulating in lungs and liver with and w/o magnetic field.  

Answer: Thank you very much for pointing out this issue. With the aim to highlight the reduction of T2 spin-spin relaxation signals due to the magnetic particle concentration (i.e., EMHV) in the animal lungs, two MRI maps - one coming from the untreated (Figure 4a) and the other one coming from the EMHVs treated mice (Figure 4b) – were included in Figure 4. Details about relaxation signals are also reported in the text – specifically from line 260 to 274.

However, for better explain this concept, an additional paragraph was added on pag 7 of the revised paper: “In order to evaluate the EMHVs concentration in animal tissues, MRI T2-weighted images – and in particular mean and the median T2 relaxation time values, were investigated.

We added a Supplementary Figure S1 to the final manuscript to demonstrate the selective accumulation of EMHV under the influence of the magnetic field in the lung, and images of the lung without the magnetic field. We also demonstrated the T2 signal mapping perturbation in the liver (Supplementary Figure S2), where no difference was observed among naïve (a), EMHV without magnetic force (b) and EMHV under the influence of the magnetic force (c). These suggest that no hepatic accumulation related to EMHV denaturation was present with or without application of the magnetic field on the lung region.

4.The effect on biocompatibility of carrier RBC must be addressed more rigorously both in vitro and in vivo. Initial naïve RBC and resultant RBC-DDDs must be tested side-by-side for sensitivity of RBC to osmotic lysis, complement, mechanical stress and oxidative stress, as described and in vivo by direct tracing of blood clearance and uptake by RES, lungs, kidneys (PMID 27003833; 29371620).

Answer: We thanks the referee for the suggestion and we are planning to perform direct tracing of blood clearance and uptake by RES, lungs, kidney in future in in vivo experiments to assess better the biodistribution. As regard the biocompatibility of EMHV carriers it is already been assessed in Taranta, M.; Naldi, I.; Grimaldi, S.; Salvini, L.; Claudio, P.P.; Rocchio, F.; Monuz, A.F.; Prete, S.; Cinti, C. Magnetically Driven Bio reactors as new Tools in Drug Delivery. J. Bioanal. Biomed. 2011, S5, doi:10.4172/1948-593X.S5-002.

5.Key parameters of design (size, PDI, charge, content) need to be indicated. What is function of FHA in this study? How many FHA units are per RBC? Which are “the targeted cells” (line 312), and why FHA-RBC supposed to selectively target them?

Answer: We have improved the original manuscript stressing that FHA is a fusogenic protein that allows a better anchoring of the EMHV element at the host cell membrane. This interaction is also favoured by the effect of the magnetic field that we believe holds EMHV in place driving membrane fusion and in turn content releasing. Detailed mechanism of EMHV interaction with the host cell membrane is already described in Cinti, C.; Taranta, M.; Naldi, I.; Grimaldi, S. Newly engineered magnetic erythrocytes for sustained and targeted delivery of anti-cancer therapeutic compounds. PLoS One 2011, 6, e17132, doi:10.1371/journal.pone.0017132. We also opted for “targeted tissue district”, changing the previous concept of targeted cells that was misleading as not proven by now. However we believe that in the future, EMHV could be endowed with other recognition elements, receptor protein inter-actors, to achieve cell targeting.

  1. Informative value of tissue sections in figure 6 is low because of poor resolution and tissue characterization. Why the uptake in the sections of different thickness varies so dramatically?

Answer: We have replaced the original figure 6 in the manuscript with an improved version of the same. The new figure 6 is now fully showing the three axial images of the CT scan analysis (fig 6 a). Our intent is now to stress the localised concentration of iron signal in only specific areas of the lungs. Those areas are under the influence of the magnetic field and lay at 500 mm up to near 700mm. Outside those areas no accumulation is visible (1100 mm).

We have also improved Fig 6 b, including low and high magnification of the areas of positive reactivity for Iron and FHA epitopes. In supplementary we also included Supplementary Figure S3 to show lack of positive signal of iron particles at 1100 mm outside the area of magnetic field influence.

Minor issues

Discussion is too long. Generally, all text and literature related to the respiratory route can be removed.

Answer: We revised the discussion and references as suggested.

Reviewer 2 Report

The authors present an interesting development and characterization study of an erythrocyte-derived delivery system with potential in drug lung targeting,  but some major modifications are required before this manuscript acceptance.

In lines 397-399 there is a typographical error because instructions on the composition of the manuscript appear. From a general point of view, the material and methods section is excessively long and too detailed. It is obviously necessary, but it should summarize the most important methodological information. The authors should make a synthesis effort to make this section of the manuscript more understandable and readable, according to the usual standards in scientific publications.

Regarding the scanning conductance microscopy (SICM) results, the observed indentation is higher for VEMs and all intermediate variations (L/R, NHA and NP30), but the authors do not discuss the possible reason for this. Further explanation of these results should be included in the discussion.

In addition, there is additional information that remains unclear.

How is a magnetic field expected to be applied in the clinic? It should be justified in the text how and why the simple application of a permanent external magnet on the skin of animals could serve as a model.

On the other hand, in the experimental procedure 30 minutes of exposure to the local magnetic field have been applied. Have other exposure times been studied? If so, this information should be included and, if not, the chosen time should be justified.

Author Response

The authors present an interesting development and characterization study of an erythrocyte-derived delivery system with potential in drug lung targeting, but some major modifications are required before this manuscript acceptance. 

  1. In lines 397-399 there is a typographical error because instructions on the composition of the manuscript appear. From a general point of view, the material and methods section is excessively long and too detailed. It is obviously necessary, but it should summarize the most important methodological information. The authors should make a synthesis effort to make this section of the manuscript more understandable and readable, according to the usual standards in scientific publications.

Answer: We thank Referee 2 for the comments. We have deleted the typographical error in the lines 397-399. The part describing the SICM setup and measurement protocols has been shortened as well as other parts in the materials and methods.

  1. Regarding the scanning conductance microscopy (SICM) results, the observed indentation is higher for VEMs and all intermediate variations (L/R, NHA and NP30), but the authors do not discuss the possible reason for this. Further explanation of these results should be included in the discussion. in addition, there is additional information that remains unclear.

Answer: In the results and in the discussion sections we added sentences providing the possible reason for the observations.

  1. How is a magnetic field expected to be applied in the clinic? It should be justified in the text how and why the simple application of a permanent external magnet on the skin of animals could serve as a model. On the other hand, in the experimental procedure 30 minutes of exposure to the local magnetic field have been applied. Have other exposure times been studied? If so, this information should be included and, if not, the chosen time should be justified.

Answer: Magnetic fields are coming popular in minimally invasive surgery, especially conceived as an innovative technical solution for reducing the number of incisions and thus for limiting problems related to the surgical trauma and/or post-intervention complications. Just as an example high interest is growing around the idea to have magnetic instruments that are inserted laparoscopically through an entry in the peritoneal cavity at one point and then driven into position elsewhere and controlled with external magnets – some references are: [Best SL, Cadeddu JA. Development of magnetic anchoring and guidance systems for minimally invasive surgery. Indian J Urol. 2010 Jul;26(3):418-22. doi: 10.4103/0970-1591.70585. PMID: 21116365; PMCID: PMC2978445], [M Brancadoro, S Tognarelli, G Ciuti, A Menciassi. A novel magnetic-driven tissue retraction device for minimally invasive surgery, Minimally Invasive Therapy & Allied Technologies, 2017].

Proposing the same working principle reported in literature, i.e., by using an external magnet properly positioned and held in the right place during the treatment, no major issues were raised during the discussion with clinicians and also literature is continuously pointing out the growing interest around the using of magnetic field and magnetic forces as a valid tool for non-invasive treatment.

We reported in the paper the first experimental protocol used just for demonstrating the potentialities of using a simple permanent external magnet held on the skin of an animals for guiding the EMHV concentration in a specific anatomical district. The results of this study should serve as a reference for the future investigations.

For better presenting these issues in the text, we added details about external magnet positioning and literature interest a little everywhere in the revised text.

Timing of the experiment was selected to comply with 3R ARRIVE guidelines for preserving the animal wellbeing. 30 min appeared as an effective time to achieve clear preliminary results, without stressors. On the other hand, in our case here, the EMHV did not contain any therapeutic molecule and did not deliver therapies. Eventually, once EMHV will be loaded with active molecules, a full characterisation will be conducted on timing of magnet application and dosing.

Reviewer 3 Report

1. The author uses MRI (T2 relaxation time) to evaluate the delivery of EMHV encoporated with specific magnetic nanoparticles (super paramagnetic iron oxide nanoparticles: SPICON) into the lung in vivo. However, the presentation of the results is really difficulty to understand for the specialist, firstly, the content mentioned ROI (region of interest), but I can't recognize where is ROI in the Figure 4.  Because I can't recognize what is the hypointense signals in the Figure 4, how to qualify the hypointense signals is evidence in EMHV treated lung?

2. For immunostaining, iron oxide nanoparticle incoporated with EMHA is stained with Perl's Prussion Blue and FHA fused protein in the membranes of EMHA is stained with anti-FHA DAB staining. These two staining (Perl's Prussion Blue and anti-FHA DAB staining in the manuscript approved both iron oxide nanoparticle and EMHA are present in the lung. However, these results can't recognize EMHA and nanoparticle are presented in the same regions of the lung, and it is possible that nanoparticle and EMHA into the lung independently. In the summary, the results of Figure 6 can't prove nanoparticle integrated with EMHA into the lung tissue.

3. The description of specialized terms in the figure 3, 4, and 6 should be more clear and qualified.

4. Moderate English change are required.

Author Response

  1. The author uses MRI (T2 relaxation time) to evaluate the delivery of EMHV encoporated with specific magnetic nanoparticles (super paramagnetic iron oxide nanoparticles: SPICON) into the lung in vivo. However, the presentation of the results is really difficulty to understand for the specialist, firstly, the content mentioned ROI (region of interest), but I can't recognize where is ROI in the Figure 4.  Because I can't recognize what is the hypointense signals in the Figure 4, how to qualify the hypointense signals is evidence in EMHV treated lung?

Answer: We thanks the Referee 3 for the comments. Figure 4 represents the ROI of the analysis. From the animal MRI images, we selected the region of interest, i.e., lung or liver (as in Figure3), and we used them as a reference for the entire analysis. Better references were included in the revised text of the manuscript.

Regarding the hypo-intense signals, if you compare the colour maps of Figure 4b and Figure4a, you can see different T2 relaxation time values that correspond to a different colour in the pixels of the maps. Specific “mean and median T2 relaxation time values” were calculated by using a dedicated algorithm in the software of the MRI machine.

  1. For immunostaining, iron oxide nanoparticle incoporated with EMHA is stained with Perl's Prussion Blue and FHA fused protein in the membranes of EMHA is stained with anti-FHA DAB staining. These two staining (Perl's Prussion Blue and anti-FHA DAB staining in the manuscript approved both iron oxide nanoparticle and EMHA are present in the lung. However, these results can't recognize EMHA and nanoparticle are presented in the same regions of the lung, and it is possible that nanoparticle and EMHA into the lung independently. In the summary, the results of Figure 6 can't prove nanoparticle integrated with EMHA into the lung tissue.

Answer: We have improved figure 6 in the manuscript and added a supplementary information Figure S3 Figure 6 now shows with no ambiguity that the slices used for the Perl’s and for the FHA staining were consecutive slices. We could not perform double staining, because of the incompatible protocols. Finally, the simultaneous presence of positive Perl’s staining ( iron) and FHA reactivity strongly imply that EMHV have been directed , anchored and fused locally as FHA is unique component of EMHV and not endogenous to the mouse and that Irons particles were included in EMHV.

  1. The description of specialized terms in the figure 3, 4, and 6 should be more clear and qualified.

Answer: Thanks. We revised it

  1. Moderate English change are required.

Answer: Thanks we revised it.

Reviewer 4 Report

Submitted manuscript entitled "Erythro-Magneto-HA-Virosome: a bio-inspired drug delivery 2 system for active targeting of drugs in the lungs" is continuation of the authors work "Lucarini, G.; Sbaraglia, F.; Vizzoca, A.; Cinti, C.; Ricotti, L.; Menciassi, A. Design of an innovative platform for the treatment of 759 cerebral tumors by means of erythro-magneto-HA-virosomes. Biomed. Phys. Eng. Express 2020, 6, 045005", which is again very successful and technically well performed, therefore we recommend accepting it in the present form.

In future development maybe the name of nanostructure “Erythro-Magneto-HA-Virosome” would be better to improve, something like magnetovirosome, would be more useful.

Author Response

Submitted manuscript entitled "Erythro-Magneto-HA-Virosome: a bio-inspired drug delivery 2 system for active targeting of drugs in the lungs" is continuation of the authors work "Lucarini, G.; Sbaraglia, F.; Vizzoca, A.; Cinti, C.; Ricotti, L.; Menciassi, A. Design of an innovative platform for the treatment of 759 cerebral tumors by means of erythro-magneto-HA-virosomes. Biomed. Phys. Eng. Express 2020, 6, 045005", which is again very successful and technically well performed, therefore we recommend accepting it in the present form.

  1. In future development maybe the name of nanostructure “Erythro-Magneto-HA-Virosome” would be better to improve, something like magnetovirosome, would be more useful.

Answer: We gratefully thanks Referee 4 for the concern, however we would like to point out that the name belongs to a Patented element and that the acronyms EMHV is often used to simplify the matter.